# Validity of Wearable Gait Analysis System for Measuring Lower-Limb Kinematics during Timed Up and Go Test

**DOI:** 10.3390/s24196296

**Published:** 2024-09-29

**Authors:** Yoshiaki Kataoka, Tomoya Ishida, Satoshi Osuka, Ryo Takeda, Shigeru Tadano, Satoshi Yamada, Harukazu Tohyama

**Affiliations:** 1Faculty of Health Sciences, Hokkaido University, Sapporo 060-0812, Japan; aquarius_plus_150g@yahoo.co.jp (Y.K.); pt-osuka@huhp.hokudai.ac.jp (S.O.); tadano@wb3.so-net.ne.jp (S.T.); tohyama@med.hokudai.ac.jp (H.T.); 2Department of Rehabilitation, Health Sciences University of Hokkaido Hospital, Sapporo 002-8072, Japan; 3Faculty of Engineering, Hokkaido University, Sapporo 060-8628, Japan; r.takeda@eng.hokudai.ac.jp (R.T.); syamada@eng.hokudai.ac.jp (S.Y.)

**Keywords:** timed up and go test, wearable sensor, kinematics, motion analysis, validation

## Abstract

Few studies have reported on the validity of a sensor-based lower-limb kinematics evaluation during the timed up and go (TUG) test. This study aimed to determine the validity of a wearable gait sensor system for measuring lower-limb kinematics during the TUG test. Ten young healthy participants were enrolled, and lower-limb kinematics during the TUG test were assessed using a wearable gait sensor system and a standard optical motion analysis system. The angular velocities of the hip, knee, and ankle joints in sit-to-stand and turn-to-sit phases were significantly correlated between the two motion analysis systems (*R* = 0.612–0.937). The peak angles and ranges of motion of hip, knee, and ankle joints in the walking-out and walking-in phases were also correlated in both systems (*R* = 0.528–0.924). These results indicate that the wearable gait sensor system is useful for evaluating lower-limb kinematics not only during gait, but also during the TUG test.

## 1. Introduction

The number of elderly people is increasing worldwide, and the rise in medical and nursing care costs is a social problem that must be addressed immediately [1]. As the population ages, the number of people suffering from musculoskeletal disorders is increasing, and the decline in quality of life due to reduced mobility has become a concern [2].

The timed up and go (TUG) test is used to assess combined mobility in daily life. This test is a modification of the get-up-and-go test by Podsiadlo et al. [3], measuring the time required to get up from a chair, walk 3 m, change direction, and sit back down in the chair at the starting point. The TUG test assesses multiple abilities of movement, including standing up, turning, and sitting down, in addition to walking. Although the TUG test completion time is used worldwide as an indicator of fall risk [4], it is difficult to determine which specific movement causes a problem based solely on the completion time. The completion time does not indicate disease-specific problems [5].

Salarian et al. [5] recently assessed patients with Parkinson’s disease using the TUG test, which includes five phases: sit-to-stand, walking-out, turning, walking-in, and stand-to-sit. Phase-based assessments can evaluate not only walking but also standing up, sitting down, and turning, which are closely related to daily life activities. In addition, some studies have shown that motion analysis during the TUG test in patients with Parkinson’s disease is reliable [6], and phase-based assessment can detect movement patterns in each patient [7]. Recent studies have evaluated lower-limb kinematics in subtasks during the TUG test [8,9,10,11]. Frail older adults take longer to stand up and show less trunk flexion during the sit-to-stand phase [8]. Furthermore, another study has shown that subjects with locomotive syndrome have smaller angular velocities of the lower-limb joints during the sit-to-stand and stand-to-sit phases [11]. Therefore, it is important to evaluate lower-limb kinematics during subtasks, including the sit-to-stand and stand-to-sit phases, to assess movement patterns during the TUG test in detail.

The gold standard for evaluating lower-limb kinematics during the TUG test is an optical motion analysis system that uses reflective markers. However, such a system requires dedicated space for measurement and considerable training. Therefore, it is necessary to develop and apply alternative motion analysis systems to overcome these limitations in large-scale surveys of older adults and in evaluations of patients with musculoskeletal disorders.

In recent years, motion analysis during the TUG test using acceleration and gyro sensors has been developed [12,13,14]. The advantage of a sensor-based motion analysis system is that it is compact, easily wearable, and transportable. Additionally, this system offers valuable information under realistic conditions. Some studies have validated sensor-based motion analysis during the TUG test [15,16]. We have also developed the H-Gait system, a motion analysis system using acceleration and gyro sensors [17,18]. Furthermore, this system has been clinically applied to individuals with knee osteoarthritis [19], hip osteoarthritis [20], and locomotive syndrome [11,21]. The advantages of this system include the ability to measure over long distances and the ability to measure in various environments, such as a positive-pressure treadmill where markers cannot be tracked by cameras [20,22], and a local community center instead of a laboratory [11,21]. Compared with conventional wearable sensor systems, this system has unique advantages, such as drift removal via the periodicity of gait and calculation of lower-limb kinematics using gravity vectors [23,24,25,26]. Tadano et al. [24] demonstrated a strong correlation between the H-Gait system and a standard optical motion analysis system in assessing lower-limb kinematics during gait. However, because the H-Gait system uses the periodicity of gait to remove drift, it is unclear whether it can be applied to the TUG test, which includes sit-to-stand and stand-to-sit movements. Therefore, the purpose of this study was to compare the validity of the H-Gait system with that of a standard optical motion analysis system in assessing lower-limb kinematics during the TUG test.

## 2. Materials and Methods

### 2.1. Participants

Ten young healthy subjects (5 male and 5 female participants; age: 23.3 ± 1.5 years; body height: 165.8 ± 9.3 cm; body weight: 57.4 ± 8.2 kg) participated. Individuals without musculoskeletal or neuromuscular disorders affecting TUG test performance were enrolled. The study protocol was approved by the Institutional Review Board of the Faculty of Health Sciences, Hokkaido University (#24-9). Written informed consent was obtained from all participants.

### 2.2. TUG Test

All the participants performed the TUG test at a self-selected speed. Lower-limb kinematic data were collected simultaneously via two independent motion analysis systems: a sensor-based system (H-Gait system) and a standard marker-based optical motion analysis system. Participants were instructed to stand up from a chair without armrests (height: 45 cm), walk 3 m, turn around a cone, walk back to the chair, and sit down while turning. For assessment via the H-Gait system, participants were asked to maintain static standing before and after the TUG test [11]. After one practice trial for each person, two trials, one with a right turn and one with a left turn, were recorded in random order.

### 2.3. Assesment via the H-Gait System

Seven sensor units (TSDN121; ATR-Promotions, Inc., Kyoto, Japan) were attached to the following body segments: pelvis, bilateral thighs, shanks, and feet. Specifically, the pelvic sensor was attached at the level of the iliac crest, the thigh sensors were at the mid-thigh, the shank sensors were at the mid-shank, and the foot sensors were at the mid-foot (Figure 1). The sensors consisted of triaxial acceleration and gyro sensors, and the sampling rate was set at 100 Hz.

The data were analyzed using a customized MATLAB 2019a program (MathWorks Inc., Natick, MA, USA). The wearable sensors were calibrated to convert the sensor coordinate system to the body segment coordinate system [24]. An initial calibration was performed to obtain the rotation matrix to convert the sensor to the global coordinate system. This process calculates the difference in the inclination of each sensor relative to gravity via acceleration data between the standing (upright) and sitting (inclined) positions (Figure 2). In previous studies of elderly individuals with musculoskeletal disorders [11,20], if participants complained of some difficulty in the sitting (inclined) position, calibration was performed using a chair with a backrest. In this study, no participants experienced such difficulty. Assuming that the sensors were located in the sagittal plane, the sensor coordinate system was transformed to the global coordinate system. This process also minimizes errors due to attachment misplacement [11]. Second, the global coordinate system was transformed into a body segment coordinate system. Retroreflective markers were applied to the bilateral greater trochanters, medial/lateral femoral epicondyles, and medial/lateral malleoli. The inclination of each segment relative to the global coordinate system was measured using three images (front, right, and left sides), and then a rotation matrix was calculated to convert the global coordinate system to the body segment coordinate system. Finally, the sensor coordinate system was transformed into the body segment coordinate system via these two rotation matrices (the sensor-to-global and the global-to-body conversions).

To construct a wire-framed human model, the following anthropometric measurements were obtained via a tape measure: hip width (distance between the bilateral greater trochanters), thigh length (distance between the greater trochanter and lateral femoral condyle), shank length (distance between the lateral femoral condyle and lateral malleolus), and foot height (from the floor to the lateral malleolus). Based on the wire-frame model, lower-limb posture was quantified during the TUG test.

### 2.4. Assessment via a Standard Optical Motion Analysis System

A total of 40 retroreflective markers were attached to each participant. Twenty-four markers were placed on the following landmarks (Figure 1): the iliac crest, anterior/posterior superior iliac spine, medial/lateral femoral epicondyles, medial/lateral ankles, medial aspect of the first metatarsal head, second metatarsal head, fifth metatarsal head, and heel, as described in a previous study [27]. The greater trochanter markers were not used for the analysis of the optical system. Additionally, the marker clusters were attached to the thigh and shank (four markers each). Marker trajectory data were obtained using a standard three-dimensional motion analysis system (Cortex version 5.0.1; Motion Analysis Corporation, Rohnert Park, CA, USA) equipped with seven near-infrared cameras (Hawk cameras; Motion Analysis Corporation, Rohnert Park, CA, USA) at a sampling rate of 200 Hz.

Kinematic analysis was performed via Visual3D version 6 (HAS-Motion, Inc., Kingston, ON, Canada). A fourth-order low-pass Butterworth filter with a cutoff frequency of 12 Hz was used to remove noise from the marker trajectories. A lower-limb skeletal model consisting of the pelvis, thighs, shank, and foot segments was constructed for each participant, and the lower-limb kinematics were calculated. The coordinate system for each segment was a right-handed Cartesian coordinate system, with the X-axis (mediolateral) oriented to the right, the Y-axis (anteroposterior) oriented forward, and the Z-axis (inferior-superior) oriented upward. A joint coordinate system with the Cardan XYZ rotational sequence was used to calculate the joint angles (flexion/extension and dorsiflexion/plantarflexion were the first). Positive values represent flexion and dorsiflexion. The degree of each joint angle was set to zero during the static standing trial, similar to the H-Gait system.

### 2.5. Phase-Based Analysis of TUG

The phase-based analysis of the TUG test included the following five phases: sit-to-stand, walking-out, turning, walking-in, and turn-to-sit [14]. Phase classification was based on pelvis and thigh gyro data for sensor-based analysis [28] (Figure 3). In the optical method, the angular velocities of the pelvis and thigh segments relative to the global coordinate system were used. The first phase, sit-to-stand, begins when the pelvic angular velocity in the roll direction (sagittal plane) exceeds 10°/s (Figure 3a). The second phase, walking-out, begins when the angular velocity of the roll axis (sagittal plane) of either the left or right thigh exceeds 10°/s (Figure 3b). The third phase, turning, is defined as the time when the pelvic angular velocity around the pitch axis (horizontal plane) exceeds 35% of the maximum absolute value during the walking phase (Figure 3c). For the fourth phase, walking-in, the onset is defined as the time when the pelvic angular velocity in the pitch direction (horizontal plane) falls below 35% of the maximum angular velocity (Figure 3d). Finally, for the fifth phase, turn-to-sit, the onset is defined as when the angular velocity in the pitch direction of the pelvis (horizontal plane) exceeds 35% of the maximum angular velocity (Figure 3e), and ends when the roll angular velocity in the pelvis (sagittal plane) falls below 10°/s (Figure 3f).

### 2.6. Data Analysis

Right lower-limb kinematics were used in the data analysis. The values of interest were determined according to a previous study [11]. In the sit-to-stand phase, average angular velocities were calculated as the difference between peak flexion (dorsiflexion) and peak extension (plantarflexion) divided by time (Figure 4a). In the walking-out phase, peak angles were calculated (Figure 4b,c). For the walking-out and walking-in phases, analysis was performed for each gait cycle. The initial contact was determined via the angular velocity of the shank [29]. For the walking-out phase, the first gait cycle after the sit-to-stand phase was excluded from the analysis. For the walking-in phase, the first walking cycle after the turning phase was excluded from the analysis. The mean range of motion (ROM) was calculated as the difference between the peak flexion and extension (dorsiflexion and plantar flexion) of each joint (Figure 4d). For the turn-to-sit phase, average angular velocities were calculated by dividing the difference between peak extension and flexion angles by the time (Figure 4e).

### 2.7. Statistical Analysis

After normality was tested via the Shapiro–Wilk test, lower-limb kinematics were compared via Pearson’s correlation coefficient. Correlation coefficient was interpreted as follows [30]: R ≥ 0.9, extremely large; 0.7 ≤ R < 0.9, very large; 0.5 ≤ R < 0.7, large; 0.3 ≤ R < 0.5, moderate; 0.1 ≤ R < 0.3, small; and R < 0.1, trivial.

Additionally, Bland–Altman analysis was conducted to visually confirm the trend of the errors. For the Bland–Altman analysis, a scatter plot was created with the average of the values from the H-Gait system and the optical motion analysis system on the horizontal axis, and the value obtained by subtracting the results of the optical motion analysis system from those of the H-Gait system on the vertical axis. Statistical analyses were performed using SPSS Statistics 22 (IBM, Armonk, NY, USA). The statistical significance level was set at *p* < 0.05.

## 3. Results

Figure 5 shows representative waveforms of the hip, knee, and ankle joint angles during the TUG test for the optical motion analysis system and the H-Gait system. The waveforms from the two measurement systems were similar, with no obvious phase shift observed.

### 3.1. Sit-to-Stand Phase

Hip, knee, and ankle angular velocities were significantly correlated between the optical motion analysis and H-Gait systems with large to extremely large coefficients (*R* = 0.612–0.937). The hip extension angular velocity had a large coefficient (*R* = 0.612, *p* = 0.004; Figure 6a). However, a significant fixed error was detected, with greater values in the H-Gait system (mean difference: 9.8°/s, *p* = 0.026; Figure 6b). For knee extension angular velocity, a large correlation coefficient was found (*R* = 0.791, *p* < 0.001; Figure 6c), and no significant fixed error was observed (*p* = 0.120, Figure 6d). The angular velocity of ankle plantarflexion had an extremely large correlation coefficient (*R* = 0.937, *p* < 0.001; Figure 6e), and no significant bias was detected (*p* = 0.266, Figure 6f). There was no obvious difference in errors between right and left turns for all kinematics.

### 3.2. Walking-Out Phase

Significant correlations were found for all peak hip flexion/extension, knee flexion/extension, and ankle dorsiflexion/plantarflexion angles, with correlation coefficients ranging from large to extremely large (*R* = 0.528–0.924). For the hip joint, the peak flexion and extension angles had very large and large correlation coefficients (*R* = 0.791, *p* < 0.001, and *R* = 0.528, *p* = 0.017, respectively) (Figure 7a,c). Significant fixed errors were detected for both hip flexion and extension (mean difference: 5.2°, *p* < 0.001, and mean difference: 2.1°, *p* = 0.007, respectively) (Figure 7b,d). For the knee joint, the peak flexion and extension angles had extremely large and very large correlation coefficients (*R* = 0.924, *p* < 0.001 and *R* = 0.750, *p* < 0.001, respectively) (Figure 7e,g). However, significant fixed errors were found for both the peak knee flexion and extension angles, with greater errors in the H-Gait system (mean difference: 5.5°, *p* < 0.001, and mean difference: 3.0°, *p* < 0.001, respectively) (Figure 7f,h). The ankle dorsiflexion and plantarflexion had very large correlation coefficients (*R* = 0.811, *p* < 0.001; *R* = 0.850, *p* = 0.002) (Figure 7i,k). On the other hand, both the peak ankle dorsiflexion and plantarflexion angles also showed significant fixed errors, with greater errors in the H-Gait system (mean difference: 2.5°, *p* < 0.001, and mean difference: 3.1°, *p* < 0.001, respectively) (Figure 7j,l). No obvious difference in errors was observed between right and left turns.

In the Bland–Altman plot, the x-axis represents the average of the H-Gait system and the optical motion analysis system, and the y-axis represents the difference between the H-Gait system and the optical motion analysis system. The results for each subject were displayed in different colors.

### 3.3. Walking-In Phase

Hip, knee, and ankle ROMs were significantly correlated between the optical motion analysis and H-Gait systems with very large coefficients (*R* = 0.578–0.884). Hip ROM had a very large coefficient (*R* = 0.884, *p* < 0.001; Figure 8a). However, a significant fixed error was detected, with greater values in the H-gait system (mean difference: 4.7°, *p* < 0.001; Figure 8b). For knee ROM, a very large correlation coefficient was found (*R* = 0.758, *p* < 0.001; Figure 8c), and no significant fixed error was observed (Figure 8d). Ankle ROM had a very large correlation coefficient (*R* = 0.837, *p* < 0.001; Figure 8e). However, a significant fixed error was detected, with greater values in the H-Gait system (mean difference: 2.6°, *p* = 0.003, Figure 8f). There was no obvious difference in the trend of errors between right and left turns.

### 3.4. Turn-to-Sit Phase

Hip, knee, and ankle angular velocities were significantly correlated between the optical motion analysis and H-Gait systems with very large coefficients (*R* = 0.723–0.804). The hip flexion angular velocity had a very large coefficient (*R* = 0.747, *p* < 0.001; Figure 9a). However, a significant fixed error was detected, with greater values in the H-Gait system (mean difference: 7.4°/s, *p* = 0.015; Figure 9b). For the knee flexion angular velocity, a very large correlation coefficient was found (*R* = 0.804, *p* < 0.001; Figure 9c), and no significant fixed error was observed (*p* = 0.851, Figure 9d). Ankle dorsiflexion angular velocity had a very large correlation coefficient (*R* = 0.723, *p* < 0.001; Figure 9e), and no significant fixed error was detected (*p* = 0.362, Figure 9f). Right and left turns showed similar error trends.

## 4. Discussion

This study aimed to evaluate the validity of the H-Gait system, a sensor-based motion analysis system, for assessing lower-limb kinematics during the TUG test. Although some reports have compared lower-limb kinematics during walking between the H-Gait system and an optical motion analysis system [24], there are no reports on the validity of the H-Gait system in assessing the sit-to-stand and turn-to-sit phases in the TUG test. This study revealed that the kinematics of the knee and ankle joints via the H-Gait system were strongly correlated with those via the optical motion analysis system, not only in the walking phase, but also in the sit-to-stand and turn-to-sit phases.

In the sit-to-stand phase, the joint angular velocities were significantly correlated between the H-Gait system and the optical motion analysis system. The correlation coefficient for the hip extension angular velocity was large but slightly lower than that for the knee extension angular velocity and ankle plantarflexion angular velocity. The disagreements were randomly distributed among the subjects, regardless of left or right turn. This may be due to the influence of pelvis sensor attachment. In the optical motion analysis system, the angle of the hip joint was calculated as the angle of the thigh segment relative to the pelvis, whereas in the H-Gait system, the hip joint angle was calculated as the angle of the thigh sensor relative to the pelvis sensor. The pelvis sensor was attached to the level of the iliac crest. Therefore, the motion of the pelvis sensor may be affected by lumbar spine motion. The Bland–Altman plot (Figure 6) shows that the hip angular velocity of the H-Gait system tended to be greater than that of the optical motion analysis system. Improving the attachment of the pelvis sensor may reduce the difference in hip joint angular velocity between the two measurement systems. A future study should be conducted to investigate better attachment of the pelvis sensor.

In the turn-to-sit phase, the angular velocities calculated via the H-Gait system were significantly correlated with those calculated via the optical motion analysis system, with very large correlation coefficients for hip flexion, knee flexion, and ankle dorsiflexion. Compared with healthy subjects, frail elderly individuals take longer to perform standing and sitting movements [31], and subjects with locomotive syndrome have lower angular velocities of the lower-limb joints during these movements [11]. Therefore, the angular velocities of lower-limb joints during the TUG test calculated in this study are useful indicators for detecting mobility impairments in frail elderly people. In the sit-to-stand and turn-to-sit phases in the TUG test, the angular velocities of the lower-limb joints calculated by the H-Gait system showed acceptable validity; however, the angular velocity of hip extension during the sit-to-stand phase tended to be greater in the H-Gait system than in the optical motion analysis system.

In the walking-out phase, very large to extremely large correlation coefficients were found for the peak angle of knee flexion/extension and ankle dorsiflexion/plantarflexion between the H-Gait system and the optical motion analysis system. The correlation coefficient for the hip joint tended to be smaller than that for the knee and ankle joints. These results are consistent with previous reports [17,24]. Tadano et al. [24] also reported that the root mean square error was greater in the hip joint than in the knee and ankle joints, when comparing the waveforms of lower-limb joint angles between the H-Gait and optical motion analysis systems. One reason for the relatively low correlation coefficient for the hip joint between the optical motion analysis system and the H-Gait system is that the H-Gait system relies on still images and acceleration data to calculate the rotation matrix. The H-gait system is calibrated between the segment coordinate system and the sensor coordinate system on the basis that the joint angles in the standing and sitting (inclined) positions are the same. In this study, the hip angle in the sitting position may have differed from that in the standing position, which may have influenced the results. In particular, the Bland–Altman plot (Figure 7) shows that the H-Gait system resulted in a larger peak angle of hip flexion and a smaller peak angle of hip extension compared to the optical motion analysis system. Since the lower-limb kinematics of the knee and ankle joints are highly correlated, certain errors may have occurred in the rotation matrix calculation of the pelvis segment in the calibration process. It is important to perform calibration to achieve the same joint angles in both the sitting and standing positions. Consequently, a more reproducible calibration method should be investigated to improve the accuracy of evaluating hip kinematics in a future study.

In the walking-in phase, the ROMs of the hip, knee, and ankle joints showed a large correlation coefficient between the optical motion analysis and the H-Gait systems. However, the peak angles of the lower-limb joints during the turning phase could not be calculated. The H-Gait system uses the periodicity of straight walking to remove drift, but we have not been able to develop a program that can handle turn. Therefore, the H-Gait system was not able to correctly calculate the peak angles during and after turning. In the future, it will be necessary to create a program that can calculate lower-limb kinematics during and after the turning phase.

In this study, kinematic analysis during the TUG test using the H-Gait system revealed a strong correlation with the optical motion analysis system. The TUG test using the H-Gait system is easier to evaluate than using standard optical motion analysis systems. In particular, the H-Gait system can evaluate lower-limb kinematics during the TUG test in approximately 10 min without environmental constraints. To take advantage of these benefits, it would be useful to conduct the TUG test using the H-Gait system on a large scale in clinical settings and for elderly people living in communities in a variety of settings. Kataoka et al. [11] demonstrated that the TUG test using the H-Gait system in a medical checkup for 140 residents could detect locomotive syndrome more accurately than the 10-m walk test using the same system [21]. In particular, the angular velocities of the hip and knee joints during the sit-to-stand phase are significantly different between patients with stage-1 locomotive syndrome, the least severe degree of locomotive syndrome, and healthy subjects [11]. Furthermore, patients with early-stage knee osteoarthritis exhibit a smaller angular velocity of hip extension during sit-to-stand movement [32]. Therefore, using the angular velocities of the lower-limb joints during the TUG test with the H-Gait system is very useful for detecting a decrease in mobility at an earlier stage.

This study investigated the validity of lower-limb kinematics during the TUG test via a wearable sensor; however, several limitations exit. First, only 10 healthy young adults were included. In future studies, we should include a larger number of subjects, including those who are elderly or who have some movement disabilities. Second, kinematics of the lower-limb during the turning phase could not be assessed.

## 5. Conclusions

This study validated lower-limb kinematics in the TUG test via the H-Gait system against a standard optical motion analysis system. The results revealed that the kinematics of the knee and ankle joints via the H-Gait system were strongly correlated with those via the optical motion analysis system, not only in the walking phase, but also in the standing and sitting phases. However, for the peak angles of hip flexion and extension, the correlation coefficient was slightly lower than that for the peak angles of the knee and ankle joints.

## Figures and Tables

**Figure 1 sensors-24-06296-f001:**
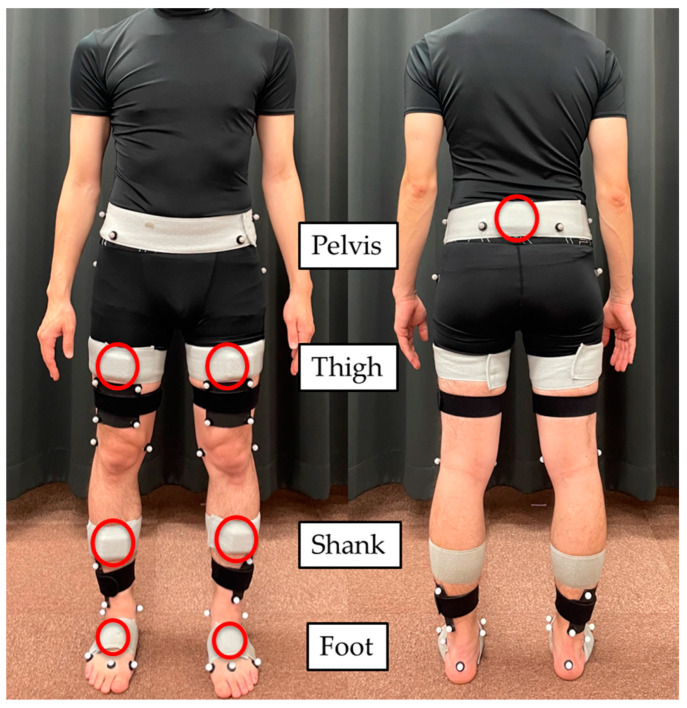
Settings of the sensor and marker placement.

**Figure 2 sensors-24-06296-f002:**
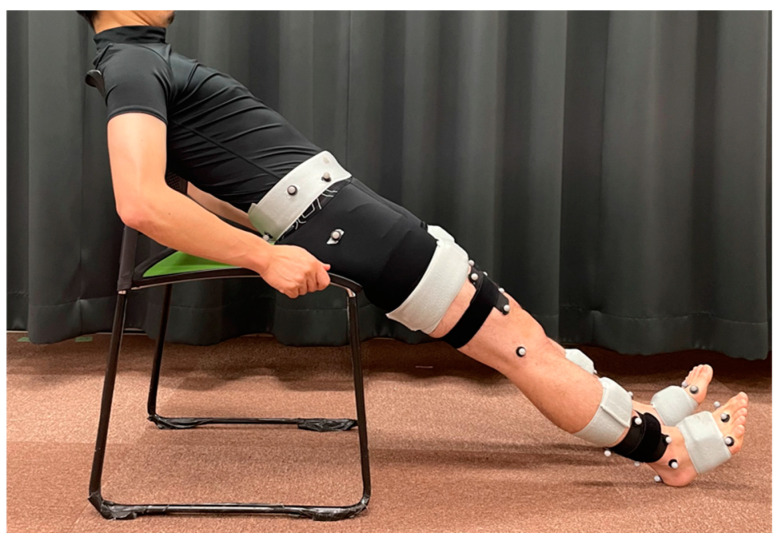
Sensor calibration at the sitting (inclined) position to convert the sensor coordinate system to the body segment coordinate system.

**Figure 3 sensors-24-06296-f003:**
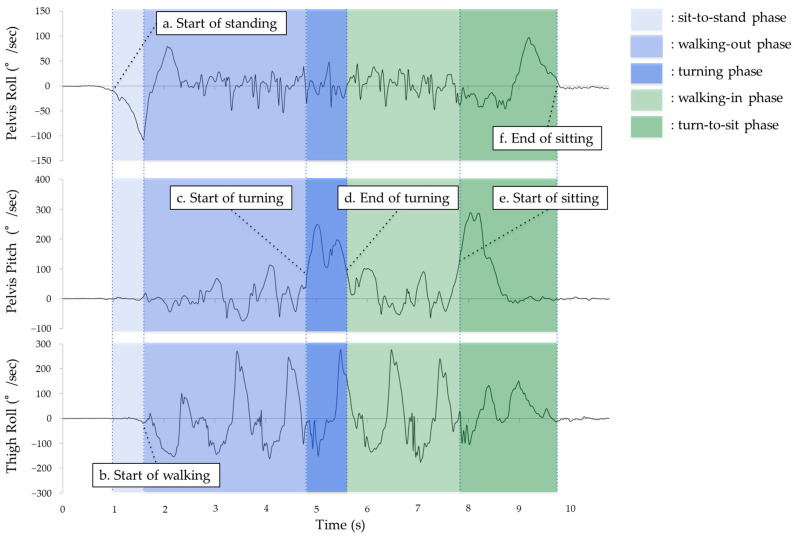
Phase classification of the TUG test using pelvis and thigh gyro sensors (the angular velocities of the pelvis and thigh). (**a**,**b**) Sit-to-stand phase, (**b**,**c**) walking-out phase, (**c**,**d**) turning phase, (**d**,**e**) walking-in phase, and (**e**,**f**) turn-to-sit phase. The roll direction represents sagittal plane motion, and the pitch direction represents horizontal plane motion.

**Figure 4 sensors-24-06296-f004:**
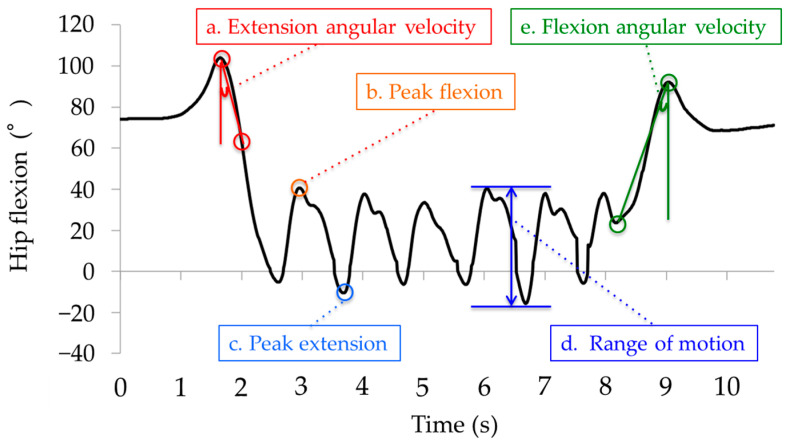
Definitions of variables (an example for hip joint). a: extension angular velocity. b: peak flexion. c: peak extension. d: range of motion. e: flexion angular velocity.

**Figure 5 sensors-24-06296-f005:**
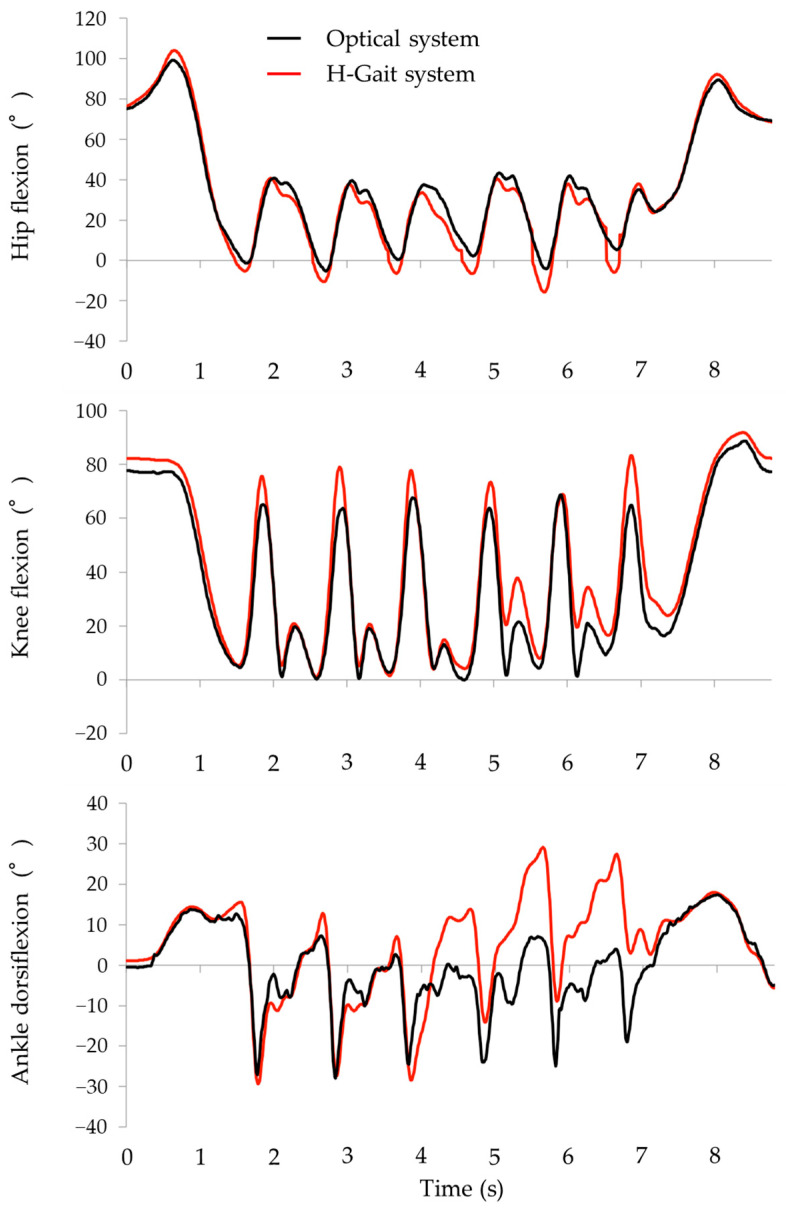
Joint angle waveforms for the hip, knee, and ankle joints during the TUG test for the optical motion analysis system and the H-Gait system (an example). Black line: optical system. Red line: H-Gait system.

**Figure 6 sensors-24-06296-f006:**
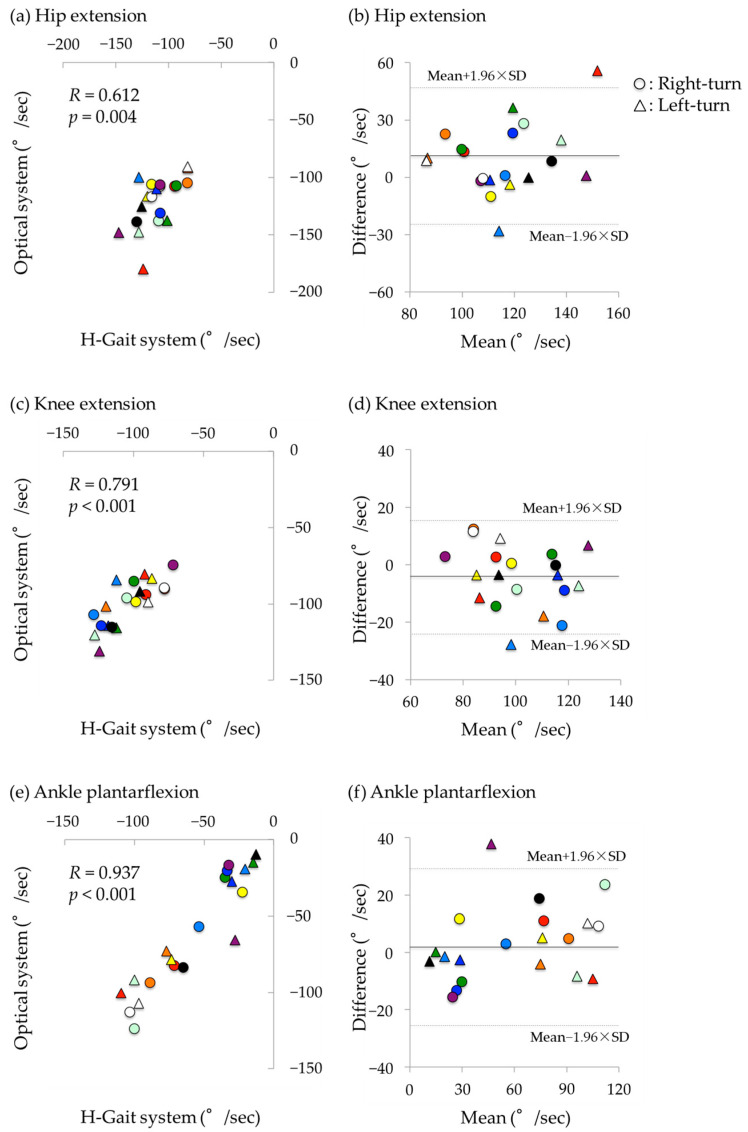
Correlation and Bland–Altman plot of angular velocities during the sit-to-stand phase between the optical motion analysis system and the H-Gait system. **Left**: Scatter plot. **Right**: Bland–Altman plot, where X-axis is the average of the H-Gait system and the optical motion analysis system, and Y-axis is the difference between the H-Gait system and the optical motion analysis system. In both plots, the results for each subject were displayed in different colors. (**a**,**b**) hip extension. (**c**,**d**) knee extension. (**e**,**f**) ankle plantarflexion.

**Figure 7 sensors-24-06296-f007:**
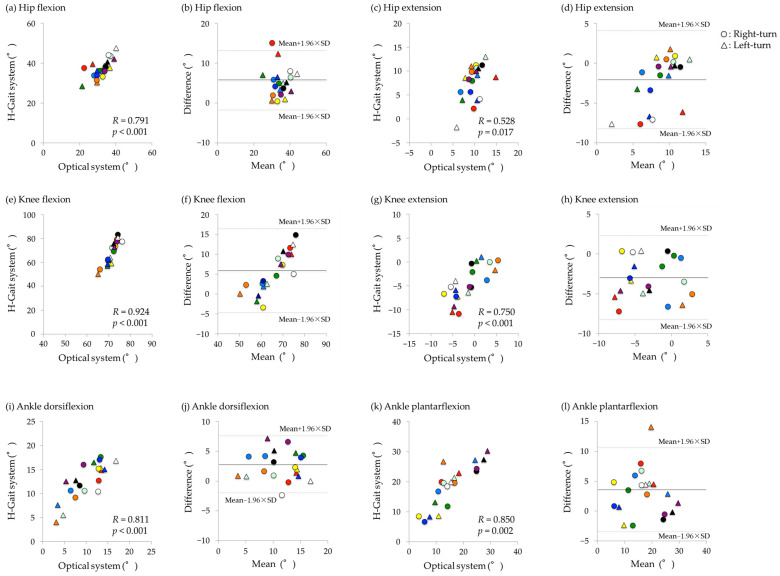
Correlations and Bland–Altman plots of peak hip, knee, and ankle joint angles during the walking-out phase between the optical motion analysis system and the H-Gait system. In both plots, the results for each subject were displayed in different colors. (**a**,**b**) hip fleion. (**c**,**d**) hip extension. (**e**,**f**) knee flexion. (**g**,**h**) knee extension. (**i**,**j**) ankle dorsiflexion. (**k**,**l**) ankle plantarflexion.

**Figure 8 sensors-24-06296-f008:**
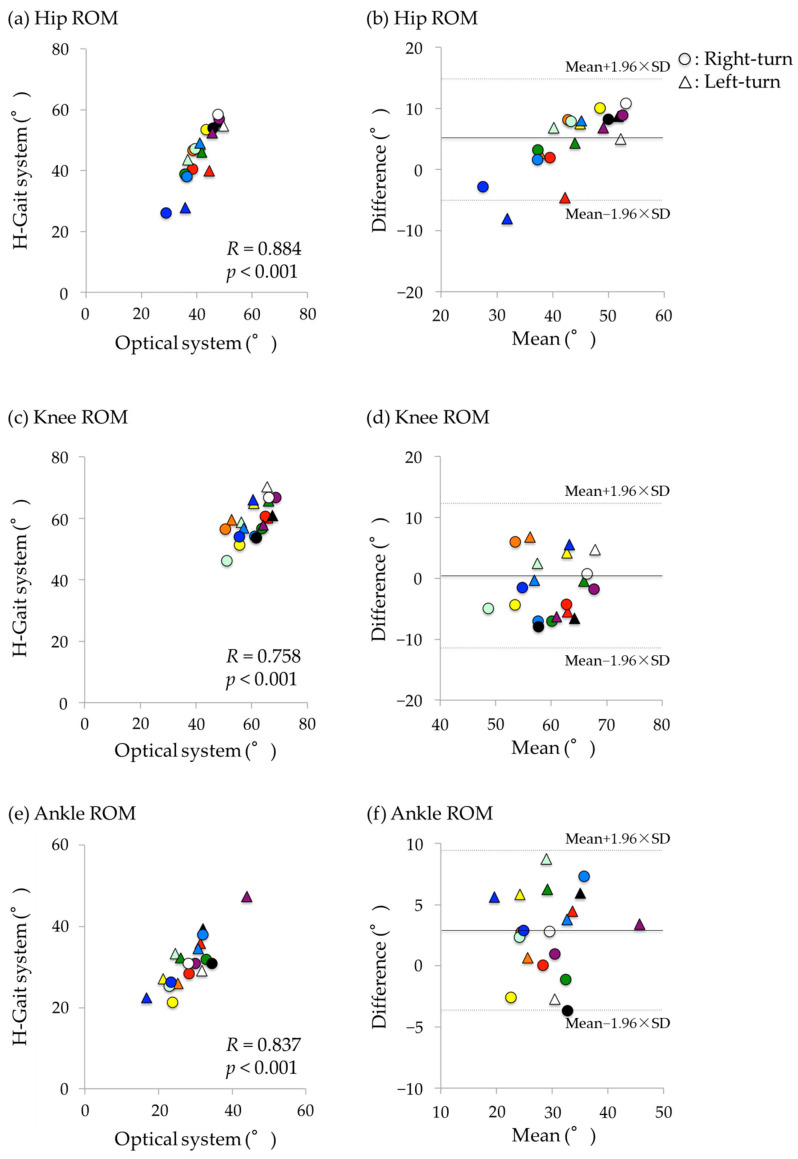
Correlations and Bland–Altman plots of the ROMs in the hip, knee, and ankle joints during the walking-in phase between the optical motion analysis system and the H-Gait system. **Left**: Scatter plot. **Right**: Bland–Altman plot, where X-axis is the average of the H-Gait system and the optical motion analysis system, and Y-axis is the difference between the H-Gait system and the optical motion analysis system. In both plots, the results for each subject were displayed in different colors. (**a**,**b**) hip ROM. (**c**,**d**) knee ROM. (**e**,**f**) ankle ROM. ROM: range of motion.

**Figure 9 sensors-24-06296-f009:**
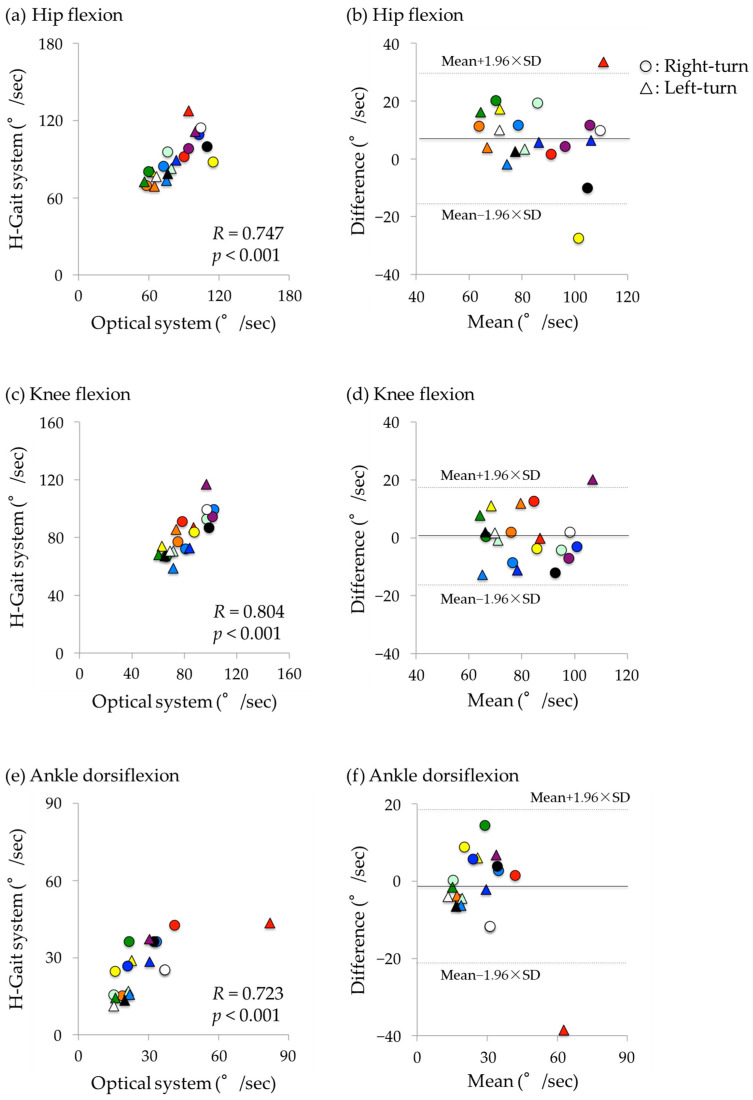
Correlation and Bland–Altman plot of angular velocities during the turn-to-sit phase between the optical motion analysis system and the H-Gait system. **Left**: Scatter plot. **Right**: Bland–Altman plot, where X-axis is the average of the H-Gait system and the optical motion analysis system, and Y-axis is the difference between the H-Gait system and the optical motion analysis system. In both plots, the results for each subject were displayed in different colors. (**a**,**b**) hip flexion. (**c**,**d**) knee flexion. (**e**,**f**) ankle dorsiflexion.

## Data Availability

The datasets used and/or analyzed during the current study are available from the corresponding author upon reasonable request.

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
