# Peer review of "Validity of Wearable Gait Analysis System for Measuring Lower-Limb Kinematics during Timed Up and Go Test"

_sensors, 2024, doi:10.3390/s24196296_

Round 1

Reviewer 1 Report

Comments and Suggestions for Authors

Thanks for the invitation to the manuscript "Validity of a Wearable Gait Analysis System for Measuring Lower-Limb Kinematics during Timed Up and Go Test" by Kataoka et al. Here are major and minor comments: 

Major comments:

> The results text for Figure 5 (lines 237 - 239) and Figure 6-13 are just a description of the figure and observations; please add some interpretation and comments on the observations; how do these high correlations mean for your objectives? These interpretations do not need to go as deep as discussion. Nevertheless, some highlights will allow the readers to understand and appreciate the results in the context of the study's objectives. 

> For the calibration phase, does it need to be done once per day? Or does it need to be performed for each participant? There might be some difficulty in clinical applications given the calibration at sitting (inclined) position on people with movement issues. 

Minor comments: 

> Line 90, why was the Review Board in "XXX". Please sync the text here with the Institutional Review Board Statement. 

> Figure 2. The first "S" was bold by mistake. 

> Pure curiosity, the definition of the b. Start of walking, it seems the changes in "Pelvis Roll" the top figure in Figure 3 undertook the largest change, and the label is on the bottom figure. Was there a reason to use the Thigh Roll instead of the Pelvis Roll?

> Figure 6 text labels are missing left bracket "(". And Figure 7 labels are missing the right bracket ")". 

> The text in Figures seems to have a minor rendering problem, where the placement of each letter was not even. 

Reviewer 2 Report

Comments and Suggestions for Authors

Comment:

This study confirms the effectiveness of H-Gait in lower limb kinematic analysis by comparing the lower limb kinematic indices of a wearable gait sensor system with those of a standard optical motion analysis system in TUG testing. The article is scholarly and readable and requires some revisions for publication in this journal:

Comments on the Quality of English Language

Please polish the language appropriately

Reviewer 3 Report

Comments and Suggestions for Authors

See attached file

Reviewer 4 Report

Comments and Suggestions for Authors

First of all, thank you very much for allowing me to review this very interesting manuscript.

Abstract

- I recommend you to define the population in the abstract, as it is a test applied in older people it can usually lead to confusion if later it is stated that the participants are healthy young adults.

Introduction

- The first sentence has no reference, I recommend that you include a demographic study that provides statistics to the reader.

- In line 39 the reference should appear after the author's name “Salarian et al. [4].”

- When talking about photogrammetry systems (gold standard) they indicate that there are complex posterior calculations, this is not totally true, because normally these systems have automatic calculations. I recommend you to focus on the importance of wearable measurements from their transportability, their easy positioning or the acquisition of data in more ecological environments.

- On line 75 the reference should appear after the author's name. Please review this issue in the entire manuscript.

Material and Method

- This section provides information in a correct and clear manner

Results

- The information provided is correct and clear. However, it would be good if the writing that appears in the different figures had a clearer and more professional format.

Discussion and Conclusions

- The information is correct

Round 2

Reviewer 3 Report

Comments and Suggestions for Authors

The authors have addressed all the comments and suggestions discussed in the first report, and introduced corresponding modifications in the manuscript.